# Genetic Variation in *CYP2D6* and *SLC22A1* Affects Amlodipine Pharmacokinetics and Safety

**DOI:** 10.3390/pharmaceutics15020404

**Published:** 2023-01-25

**Authors:** Paula Soria-Chacartegui, Pablo Zubiaur, Dolores Ochoa, Gonzalo Villapalos-García, Manuel Román, Miriam Matas, Laura Figueiredo-Tor, Gina Mejía-Abril, Sofía Calleja, Alejandro de Miguel, Marcos Navares-Gómez, Samuel Martín-Vilchez, Francisco Abad-Santos

**Affiliations:** 1Clinical Pharmacology Department, Hospital Universitario de La Princesa, Instituto Teófilo Hernando, Instituto de Investigación Sanitaria La Princesa (IP), Universidad Autónoma de Madrid (UAM), 28006 Madrid, Spain; 2Division of Clinical Pharmacology, Toxicology and Therapeutic Innovation, Children’s Mercy Research Institute, Kansas City, MO 64102, USA; 3Servicio de Bioquímica Clínica, Clínica Universidad de Navarra, 31008 Pamplona, Spain; 4Centro de Investigación Biomédica en Red de Enfermedades Hepáticas y Digestivas (CIBERehd), Instituto de Salud Carlos III, 28029 Madrid, Spain

**Keywords:** pharmacogenetics, amlodipine, CYP2D6, SLC22A1

## Abstract

Amlodipine is an antihypertensive drug with unknown pharmacogenetic biomarkers. This research is a candidate gene study that looked for associations between amlodipine pharmacokinetics and safety and pharmacogenes. Pharmacokinetic and safety data were taken from 160 volunteers from eight bioequivalence trials. In the exploratory step, 70 volunteers were genotyped for 44 polymorphisms in different pharmacogenes. CYP2D6 poor metabolizers (PMs) showed higher half-life (t_1/2_) (univariate *p*-value (*p*_uv_) = 0.039, multivariate *p*-value (*p*_mv_) = 0.013, β = −5.31, R^2^ = 0.176) compared to ultrarapid (UMs), normal (NMs) and intermediate metabolizers (IMs). *SLC22A1* rs34059508 G/A genotype was associated with higher dose/weight-corrected area under the curve (AUC_72_/DW) (*p*_uv_ = 0.025; *p*_mv_ = 0.026, β = 578.90, R^2^ = 0.060) compared to the G/G genotype. In the confirmatory step, the cohort was increased to 160 volunteers, who were genotyped for *CYP2D6*, *SLC22A1* and *CYP3A4*. In addition to the previous associations, CYP2D6 UMs showed a lower AUC_72_/DW (*p*_uv_ = 0.046, *p*_mv_ = 0.049, β = −68.80, R^2^ = 0.073) compared to NMs, IMs and PMs and the *SLC22A1* rs34059508 G/A genotype was associated with thoracic pain (*p*_uv_ = 0.038) and dizziness (*p*_uv_ = 0.038, *p*_mv_ = 0.014, log OR = 10.975). To our knowledge, this is the first work to report a strong relationship between amlodipine and *CYP2D6* and *SLC22A1*. Further research is needed to gather more evidence before its application in clinical practice.

## 1. Introduction

High blood pressure (HBP) is an important risk factor for cardiovascular morbidity and mortality [1]. HBP shows a high prevalence in the population: it affected 45.4% of adults aged 18 or over in 2017–2018 in the US, and up to 74.5% of people aged 60 or over [2]. Patients with HBP are often asymptomatic [1]. However, it is a chronic disease that increases the risk of cardiovascular, cerebrovascular and renal problems. Therefore, the European Society of Hypertension and the European Society of Cardiology (ESH/ESC) guidelines (2018) recommend patients to have their BP controlled and reduced to less than 140/90 mmHg [3]. To achieve this, an optimal control and adherence to the treatment is necessary. 

Amlodipine is a calcium channel blocker (CCB) used for the treatment of HBP [4]. Its antihypertensive effect is achieved through the vasodilatation produced by the inhibition of the transmembrane calcium influx into vascular smooth muscle cells and cardiac muscle cells [5]. Amlodipine presents an oral bioavailability of 64%. The plasma peak concentration (C_max_) is reached 6–9 h after administration (t_max_). It binds extensively to plasma proteins (98%) and shows a volume of distribution (V_d_) of 21 L/kg [6]. It is mainly metabolized by CYP3A4 and none of its metabolites are active [7]. Its elimination half-life (t_1/2_) ranges between 31 and 37 h after the administration of a single dose, with a clearance (Cl) of 0.42 L/kg*h [6]. The most frequent adverse drug reactions (ADRs) caused by amlodipine are headache, flushing, dizziness and peripheral edema, among others that are less frequent, such as hypotension and thoracic pain [6,8,9]. However, it seems to present a more favorable side effect profile and to be more cardioprotective than other antihypertensive drugs, such as metoprolol, lisinopril or irbesartan [6,10]. 

When pharmacological treatment is indicated for HBP management, the selection of the drug or drugs is usually based on individual efficacy and tolerability. However, no good response predictors are available, and it is estimated that, in Europe, only 19–40% of patients on treatment achieve target BP [3,11,12]. This lack of efficacy could be partially explained by pharmacogenetics. Genetic variation in genes coding for metabolic enzymes and transporters can alter their activity, which could modify drug exposure, therefore conditioning its efficacy and safety. To date, no clinically relevant pharmacogenetic biomarkers for amlodipine or any other antihypertensive drug have been described. Thus, our objective was to evaluate the impact of 44 single nucleotide polymorphisms (SNPs) in 15 pharmacogenes, including metabolizing enzymes (*CYP3A4, CYP3A5, CYP2D6, CYP2B6, CYP2C9, CYP2C19, CYP2C8, CYP2A6, CYP1A2* and *UGT1A1)* and transporters (*SLCO1B1, ABCB1, ABCC2, ABCG2* and *SLC22A1)* on the pharmacokinetic variability of amlodipine and ADR incidence. The present work is part of the La Princesa Multidisciplinary Initiative for the Implementation of Pharmacogenetics (PriME-PGx) [13].

## 2. Materials and Methods

### 2.1. Study Population

The participants of the present pharmacogenetic study were healthy volunteers enrolled in eight amlodipine bioequivalence clinical trials (named from A to H) conducted at the Clinical Trials Unit of Hospital Universitario de La Princesa (UECHUP), Madrid (Spain) from 2012 to 2018. The inclusion criteria were: males or females aged from 18 to 55, free from organic or psychic conditions, with normal medical records, vital signs, electrocardiogram and physical examination and without significant abnormalities in hematology, coagulation, biochemistry, serological and urine analysis. The exclusion criteria were: having received medication two days prior to the start of the study, having a body mass index (BMI) outside the 18.5–30.0 range, being pregnant or breastfeeding women, having history of sensitivity to any drug, having a positive drug screening, smoking or alcoholism, blood donation in the last month and participation in another study with investigational drugs in the three previous months. 

The EUDRA-CT numbers of the clinical trials were as follows: 2012-001846-16 (A), 2013-004147-23 (B), 2017-000547-40 (C), 2017-001716-10 (D), 2017-001757-14 (E), 2017-005024-25 (F), 2018-001378-11 (G) and 2018-002075-18 (H). All of them were approved by the Spanish Drugs Agency (AEMPS) and the Research Ethics Committee (CEIm) of the Hospital Universitario de La Princesa. Both the development of the trials and the handling of data were conducted in compliance with Spanish Legislation, the International Council on Harmonization (ICH) guidelines on Good Clinical Practice [14] and the Revised Declaration of Helsinki [15]. A total of 216 volunteers gave written informed consent to participate in the clinical trials and 160 healthy volunteers gave it for the pharmacogenetic study. 

### 2.2. Study Design and Procedures

Data were obtained from eight bioequivalence trials that compared a test and a reference formulation of amlodipine combined with other antihypertensive or hypocholesterolemic agents. A single oral dose was administered under fasting conditions in all trials, which were phase I, open-label, single-center, crossover and randomized with two sequences and two periods for amlodipine. In clinical trial A, amlodipine/atorvastatin (10/10 mg) film-coated tablets were compared with Caduet^®^ film-coated tablets (Pfizer S.A.) at the same dose. Amlodipine/valsartan (10 mg/160 mg) film-coated tablets were compared with Exforge^®^ film-coated tablets (Novartis) at the same dose in clinical trial B. In clinical trials C, D and E, amlodipine/valsartan/hydrochlorothiazide (HTZ) (10/320/25 mg) film-coated tablets were compared with Exforge HCT^®^ film-coated tablets (Novartis Europharm Limited) at the same dose. In clinical trials F, G and H, the test formulations were olmesartan/amlodipine/HTZ (40/10/12.5 mg) film-coated tablets, which were compared with Sevikar HCT^®^ (Daiichi Sankyo Spain, S.A) or with Olmetec Plus^®^ (Daiichi Sankyo Spain, S.A) and Norvas^®^ (Pfizer S.A.) tablets, both 40/10/12.5 mg. The wash-out time between the two periods was 14 days for clinical trials A to E and 21 days for clinical trials F to H. In both periods in all clinical trials, volunteers were confined at UECHUP since 10 p.m. of the day before drug administration (day 0). Drug administration took place at 9:00 a.m. (day 1), and blood samples were extracted at different times for amlodipine plasmatic quantification and for genotyping. Volunteers from clinical trials C to H stayed at UECHUP until 9:30 p.m. (day 1) and volunteers from clinical trials A and B stayed until 9:30 a.m. (day 2). All the volunteers visited UECHUP for additional blood extractions at 24, 48 and 72 h after drug administration. The studies were blinded only for the plasma concentration determination of both formulations. In all of them, the test formulation was demonstrated to be bioequivalent to the reference for amlodipine. Therefore, the arithmetic mean of the pharmacokinetic parameters of both formulations was calculated for each volunteer.

### 2.3. Pharmacokinetic Analysis

At least 16 blood samples were extracted from pre-dose to 72 h after drug intake in each period of the eight clinical trials. Amlodipine plasma concentration was measured by an external laboratory by high performance liquid chromatography with mass spectrometry (HPLC-MS/MS). The lower limit of quantification was established at 50.40 pg/mL (A), 50.25 pg/mL (B and F), 49.50 pg/mL (C, D, E and H) and 49.90 pg/mL (G). The concentration values obtained were used to calculate pharmacokinetic parameters using WinNonLin Professional Software version 2.0 (Scientific Consulting, Inc., Cary, NC, USA) for clinical trials A and B and version 7 (Scientific Consulting, Inc., Cary, NC, USA) for clinical trials C to H. The area under the curve from pre-dose to 72 h (AUC_72_) was calculated using the plasmatic concentrations according to the linear trapezoidal rule. The C_max_ and t_max_ were obtained directly from the concentration-time curve and the t_1/2_ was calculated as −ln2/K_e_.

### 2.4. Safety

Serological, biochemical and hematological tests were performed at different times during the eight clinical trials to assess safety. In addition, vital signs and BP measurements, physical examination and electrocardiograms (ECG) were scheduled at different times after drug administration. Volunteers were asked about the occurrence of adverse events (AEs) and were able to report them spontaneously. AEs causality was analyzed according to the algorithm of the Spanish pharmacovigilance system [16]. Those AEs with a possible, probable or definitive relationship with drug intake were defined as ADRs and were included in the statistical analysis.

### 2.5. Genotyping

Blood samples collected in EDTA-K_2_ tubes during the clinical trials were used to extract DNA using a MagNA Pure instrument (Roche Applied Science, USA) or a Maxwell^®^ RSC Automated DNA extractor (Promega Biotech Iberica S.L). In an initial exploratory step, 70 individuals were genotyped for 44 polymorphisms in 15 pharmacogenes. For the initial exploratory phase, genotyping was carried out in a QuantStudio 12 K Flex qPCR instrument with an OpenArray thermal block (Applied Biosystems, Thermofisher, USA) using a custom array (Table 1).

To determine the deletion (*5) or duplication of *CYP2D6,* a TaqMan copy number variation (CNV) assay (Applied Biosystems, Foster City, CA, USA) was performed with probes for exon 9 (Assay ID: Hs00010001_cn) and intron 2 (Assay ID: Hs04083572_cn) of this gene, as previously described in a published work [17]. 

Additionally, 90 healthy volunteers were genotyped for the variants in genes with significant associations in the exploratory step and for the main candidates (*CYP3A4* rs67666821 and rs35599367). For this confirmatory step, *CYP2D6* rs35742686 and rs3892097 were genotyped with qPCR using TaqMan probes (ThermoFisher IDs: C__32407232_50 and C__27102431_D0) in the pharmacogenetics unit of the Clinical Pharmacology Department of the Hospital Universitario de La Princesa in a QuantStudio 12k Flex coupled to a 96-Fast thermal block. The genotyping of *CYP2D6* rs5030656, rs1065852, rs28371706, rs28371725, *CYP3A4* rs67666821, rs355993678 and *SLC22A1* rs34059508 was conducted by MassArray (iPLEX^®^ Gold technology) at the Spanish National Genotyping Center (CEGEN- FPGMX P22-FPGMX-038) [18] and CNV genotyping was performed as previously mentioned.

### 2.6. Phenotyping and Haplotyping

Enzyme or transporter phenotypes were inferred according to the Clinical Pharmacogenetics Implementation Consortium (CPIC) or the Dutch Pharmacogenetic Working Group (DPWG) guidelines for the following genes: *CYP2B6* [19], *CYP2C19* [20], *CYP2C9* [21], *CYP2D6* [22], *CYP3A4* [23], *CYP3A5* [24], *SLCO1B1* [25] and *UGT1A1* [26]. For *CYP2C8*, *1/*1 individuals were classified as normal metabolizers (NM), *1/*2 and *1/*4 as intermediate metabolizers (IM), *4/*4 as poor metabolizers (PM), *1/*3 as rapid metabolizers (RM) and *3/*3 as ultrarapid metabolizers (UM), as previously described [27]. The SNPs genotyped in the remaining genes were analyzed individually.

### 2.7. Statistical Analysis

SPSS software (version 23, SPSS Inc., Chicago, IL, USA) was used to perform the statistical analysis. AUC_72_ and C_max_ were divided by the dose/weight ratio (DW) to correct the effect of dose and weight. Pharmacokinetic parameters were analyzed according to sex, race, clinical trial, co-administered drugs, genotypes and phenotypes. Variable distributions were checked for normality with a Shapiro–Wilks test. Variables not distributed normally were logarithmically transformed and normality was re-evaluated. For the pharmacokinetic variables following a normal distribution with two categories, a t-test was performed, whereas an ANOVA test followed by the Bonferroni post hoc was applied for those with three or more categories. For those not following a normal distribution, non-parametric tests were used. A Mann–Whitney test was used for variables with two categories and a Kruskal–Wallis test for those with three or more categories. The univariate *p*-value for statistically significant associations is shown (*p*_uv_). Those biomarkers that presented a significant association in the univariate analysis (*p*_uv_ < 0.05) were included in the multivariate analysis by means of linear regression. Significant associations are shown as the multivariate *p*-value *(p*_mv_), the non-standardized β coefficient (β) and R^2^. Additionally, a Chi^2^ or Fisher exact test was performed in the confirmatory step to evaluate the incidence of ADRs according to sex, race, clinical trial, co-administered drugs, genotypes and phenotypes. Equally, the biomarkers with a significant association in the univariate analysis (*p*_uv_ < 0.05) and the pharmacokinetic parameters AUC and C_max_ were included in the multivariate analysis, which involved logistic regression. The *p*_mv_ and the odds ratio (OR) were shown for statistically significant associations.

## 3. Results

A total of 160 volunteers participated in this study (Table 2). Women presented lower weight, height and BMI compared to men (*p* < 0.001, *p* < 0.001 and *p* = 0.002, respectively). Age was higher in self-reported Latin-Americans (*p* = 0.005) compared to self-reported Caucasians and healthy volunteers with other races (individuals who self-reported as Blacks, Asians or Arabs).

### 3.1. Pharmacokinetics

Women showed higher AUC_72_ and C_max_ compared to men (350.09 ± 811.11 h*ng/mL versus 275.88 ± 545.94 h*ng/mL, *p* < 0.001 and 10.52 ± 2.26 ng/mL versus 8.53 ± 1.59 ng/mL, *p* < 0.001, respectively), but the differences disappeared after DW correction (Table 3). Women also presented higher t_max_ (*p*_uv_ = 0.002; *p*_mv_ = 0.013, β = 0.55, R^2^ = 0.164) compared to men (Table 3). AUC_72_/DW, C_max_/DW, t_max_ and t_1/2_ were higher when an amlodipine/valsartan formulation was administered (*p*_mv_ = 0.002, β = 368.25, R^2^ = 0.073; *p*_mv_ = 0.01, β = 7.97, R^2^ = 0.036; *p*_mv_ = 0.036, β = 0.72, R^2^ = 0.164 and *p*_uv_ = 0.004, *p*_mv_ < 0.001, β = 6.04, R^2^ = 0.077, respectively) compared to the other formulations (Table 3). AUC_72_/DW was lower when atorvastatin was co-administered (*p*_uv_ = 0.018) compared to the co-administration of other drugs and t_max_ was lower in clinical trial G (*p*_uv_ = 0.007, *p*_mv_ < 0.001, β = −0.03, R^2^ = 0.164) compared to the other clinical trials. In addition, AUC_72_/DW and t_max_ were higher in Latin-Americans and healthy volunteers with other races (*p*_mv_ = 0.029, β = 195.05, R^2^ = 0.073 and *p*_uv_ = 0.022, *p*_mv_ = 0.004, β = 0.74, R^2^ = 0.164, respectively) compared to Caucasians (Table 3).

In the exploratory step, t_1/2_ was higher in CYP2D6 PMs compared to UMs + NMs + IMs (*p*_uv_ = 0.039, *p*_mv_ = 0.013, β = −5.31, R^2^ = 0.176) (Table 4). Individuals who presented the *SLC22A1* rs34059508 G/A genotype showed a higher AUC_72_/DW (*p*_uv_ = 0.025; *p*_mv_ = 0.026, β = 578.90, R^2^ = 0.060) compared to those with G/G genotype (Table 4). No pharmacokinetic differences were found for the remaining pharmacogenetic variables (Appendix A).

In the confirmatory phase, 90 additional volunteers were included, reaching a final sample size of 160. Consistently, a significantly lower AUC_72_/DW was observed in CYP2D6 UMs compared to NMs + IMs + PMs (*p*_uv_ = 0.046, *p*_mv_ = 0.049, β = −68.80, R^2^ = 0.073) and t_1/2_ was higher in PMs compared to IMs + NMs + UMs (*p_uv_* = 0.006) (Table 5, Figure 1). Volunteers with the *SLC22A1* rs34059508 G/A genotype showed a significantly higher AUC_72_/DW (*p*_uv_ = 0.046) compared to individuals with the G/G genotype, yet no significant differences in t_1/2_ were observed (Table 5, Figure 1). A tendency of higher exposure could be observed for the CYP3A4 IM phenotype. However, consistently with the exploratory step, the tendency was not significant (Table 5).

### 3.2. Safety

A total of 66 volunteers suffered from at least one ADR. The most common ADRs were headache (it affected 31.8% of volunteers), dizziness (5.63%), asthenia and eczema (3.12% each), nausea, hypotension and presyncope (2.50% each), diaphoresis (1.88%), diarrhea, cold and vomiting (1.25% each). ADR incidence was lower in men and Caucasians compared to women and to Latin-Americans and healthy volunteers with other races (*p*_uv_ = 0.024, *p*_mv_ = 0.012, log OR = 2.327; *p*_uv_ = 0.006, *p*_mv_ = 0.031, log OR = 2.306, respectively). In addition, ADR incidence was higher when a triple combination therapy was administered (*p*_uv_ = 0.011) compared to the administration of a dual therapy, and it was lower in clinical trial B (*p*_uv_ = 0.014) compared to clinical trials A + C–H (Table 6). Latin-Americans and healthy volunteers with other races also suffered from headaches with higher frequency compared to Caucasians (*p*_uv_ = 0.002, *p*_mv_ = 0.001, log OR = 3.300) (Table 6). The only individual with thoracic pain showed the *SLC22A1* rs34059508 G/A genotype (*p*_uv_ = 0.038). This genotype was also associated with a higher frequency of dizziness when compared to the G/G genotype (*p*_uv_ = 0.038, *p*_mv_ = 0.014, log OR = 10.975) (Table 6). Additionally, those individuals who suffered from dizziness showed higher AUC_72_/DW compared to those without this ADR (2406.31 ± 475.78 h*ng*kg/mL*mg versus 2107.55 ± 490.26 h*ng*kg/mL*mg, *p*_mv_ = 0.033, log OR = 1).

## 4. Discussion

Four first-line classes of drugs are used for HBP treatment, usually administered in a combination therapy: angiotensin-converting enzyme (ACE) inhibitors, angiotensin II receptor subtype 1 (AT1) blockers, long-acting CCBs of the dihydropyridine type and thiazide-like diuretics [3]. However, the high and growing incidence of HBP and the low control rates worldwide require that new strategies be developed to reach higher effectiveness of the pharmacological treatments. Pharmacogenetic biomarkers could be a useful tool to achieve the desired response. Pharmacogenetic information regarding amlodipine is scarce and no pharmacogenetic clinical guidelines are available nowadays. Hence, the present work is of great utility as it provides evidence for pharmacogenetic biomarkers and pharmacokinetic and safety variability in amlodipine treatment.

Here, amlodipine pharmacokinetics was consistent with previous works. For instance, the mean t_max_ (6.24 h) and t_1/2_ (34.89 h) fall within the ranges specified in the drug label (6–9 h for t_max_ and 31–37 h for t_1/2_) [8]. The mean AUC_72_ in this research was 312.52 ng*h/mL for a dose of 10 mg, compared to the mean AUC_t_ reported in the literature for the same dose, 238.05 ng*h/mL [28]. The concomitant use of drugs could affect the absorption and metabolism of the drug, which could explain the reportedly 24% higher AUC value.

To our knowledge, this is the first work to describe a higher amlodipine exposure and t_max_ in Latin-Americans and healthy volunteers with other races compared to Caucasians. However, pharmacokinetic differences are well known for several drugs according to race [29], which are believed to be due to the different allele frequency in the CYP P450 genes and other genes involved in drug transport and metabolism among races [30,31]. These genetic differences among races, along with the high incidence of diseases that increases the risk of pharmacological interactions and ADRs, such as diabetes or chronic kidney disease, might lead to different management of antihypertensive treatment in the Latin-American population [32,33]. Further research is needed to evaluate whether these pharmacokinetic differences have an impact on drug response and should be considered in clinical practice.

A higher amlodipine exposure was found in volunteers that were treated with amlodipine and valsartan, which corresponds to clinical trial B. The co-administration of valsartan does not seem to be the cause of the higher exposure, as it was not observed when valsartan and HTZ were co-administered in clinical trials C, D and E and no interaction between amlodipine and valsartan was described in the literature. Additionally, the higher exposure does not correlate with a higher incidence of ADRs, which suggests that these results are futile. However, it is of interest to consider them in the analysis to better understand the interactions among those variables that were considered more relevant. 

In this study, individuals with higher CYP2D6 activity showed lower amlodipine exposure and needed less time to eliminate it from the blood. To our knowledge, amlodipine is not known to be a CYP2D6 substrate, unlike diltiazem, another CCB [34]. Previous studies suggested that a fraction of amlodipine is metabolized by CYP enzymes [35], and, according to our data, one of these enzymes could be CYP2D6. The presence of a primary amine group in amlodipine supports this hypothesis, as CYP2D6 shows preference for metabolizing amines [36,37]. Additional in vitro works are required to demonstrate this enzyme–drug interaction, as well as in vivo studies to determine the clinical relevance of CYP2D6 phenotype. 

The organic cationic transporter (OCT1), encoded by the *SLC22A1* gene, is one of the major hepatic transporters that uptakes diverse drugs from blood into hepatic cells. The *SLC22A1* rs34059508 A allele was previously defined as a no function allele [38]. In this study, individuals with the *SLC22A1* rs34059508 A allele showed higher drug exposure, which is consistent with different studies with other drugs [38]. No pharmacogenetic information regarding *SLC22A1* and amlodipine is known, but it was proposed that CCBs are OCT1’s ligands [39]. According to this, amlodipine could be an OCT1 substrate: *SLC22A1* rs34059508 A allele carriers would have a reduced amlodipine uptake into hepatic cells, reducing its metabolism and, consequently, increasing drug exposure. If confirmed in further studies, variants in *SLC22A1* may become biomarkers of great relevance in amlodipine treatment.

CYP3A4 is the main enzyme involved in amlodipine metabolism. However, only a non-significant trend of higher exposure in IMs could be observed, without significant differences in safety. To our knowledge, the only study in the literature that analyzed the same *CYP3A4* variants found similar results [40]. Other studies with amlodipine focused on *CYP3A4**1B, *1G and rs2246709, and the results found were contradictory [41,42,43,44,45]. The lack of correlation between the CYP3A4 phenotype and drug exposure or response could be explained by the high inducibility and inhibition of this enzyme, which might limit its value as a pharmacogenetic biomarker for amlodipine and other drugs [46]. However, the lack of correlation might also be explained by the absence of CYP3A4 PMs; thus, it would be of great interest to analyze whether these results are confirmed when including individuals with this phenotype. CYP3A4 PMs might show higher amlodipine concentrations, as it happens when it is administered in the presence of CYP3A4 inhibitors, such as azole antifungal agents or ritonavir [47]. 

Regarding safety, the incidence of ADRs was higher in women compared to men, which is consistent with previous works, where a higher response in women was also observed [35,45,48]. The higher exposure found in women before DW correction, along with the different body mass composition and the different hormone profile, may be responsible for the safety differences between men and women [48,49]. The incidence of ADRs was also higher when volunteers were given a triple combination therapy instead of a dual therapy, as was expected, since the administration of a higher number of drugs increases the risk of suffering from an ADR. According to our data, a previous study suggested that dual therapy is safer than triple therapy [50], but these results were not replicated in other works [51,52]. Lastly, Latin-Americans and healthy volunteers with other races suffered from ADRs with higher frequency compared to Caucasians, which is consistent with the higher amlodipine exposure observed in these individuals and also with the results of previous research [53]. Additional studies are required to clarify the clinical relevance of these associations and their utility in clinical practice. The only SNP significantly related to the higher incidence of ADRs was *SLC22A1* rs3459508. Individuals who showed the rs3459508 G/A genotype suffered from dizziness and thoracic pain with higher frequency and also showed higher amlodipine exposure. Therefore, safety results reinforce the hypothesis that amlodipine is an OCT1 substrate, as those individuals with lower transporter activity show higher amlodipine exposure and have higher risk of suffering from ADRs.

The main limitation of this research was the administration of a single dose to healthy volunteers, which prevented us from analyzing amlodipine effectiveness or long-term safety. In addition, *CYP3A4* could not be properly assessed, as no PMs were identified. By contrast, it should be noticed that the controlled environment in which the study was conducted allowed us to avoid cofounding factors, such as concomitant medications or smoking. Furthermore, additional research, such as more candidate gene studies and genome-wide association studies (GWAS), could provide unknown pharmacogenetic biomarkers involved in amlodipine pharmacokinetics or pharmacodynamics, which could also play an important role in predicting the response to this drug.

## 5. Conclusions

The CYP2D6 phenotype conditioned amlodipine exposure, which suggests it plays an important role in its metabolism. Furthermore, *SLC22A1* affected amlodipine exposure and safety, which suggests it is involved in its transport. To the best of our knowledge, this is the first work to find these associations and to propose *CYP2D6* and *SLC22A1* as potential pharmacogenetic biomarkers in amlodipine treatment. Even though both associations are robust, their novelty and the absence of information in the literature make it necessary to gather more evidence before implementing their genotyping in routine clinical practice.

## Figures and Tables

**Figure 1 pharmaceutics-15-00404-f001:**
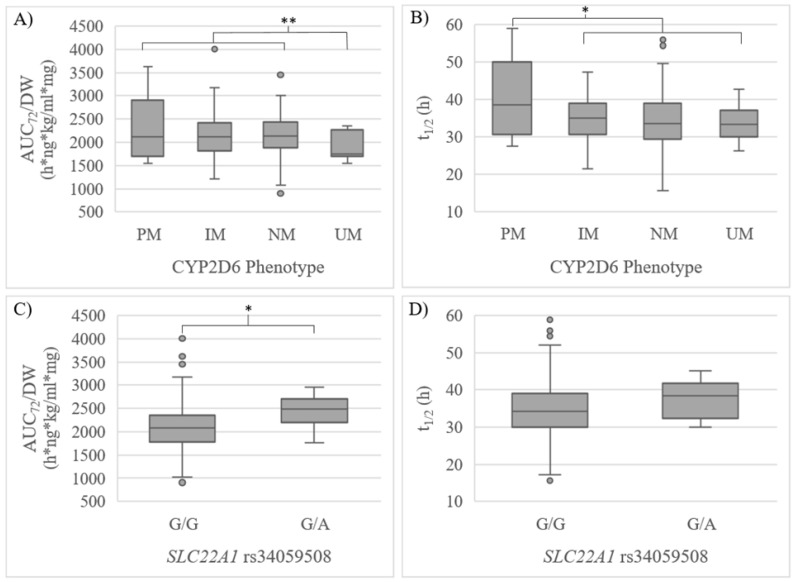
(**A**) AUC_72_/DW and (**B**) t_1/2_ regarding CYP2D6 phenotype; (**C**) AUC_72_/DW and (**D**) t_1/2_ regarding *SLC22A1* rs34059508 in the confirmatory step. UM: ultrarapid metabolizer, NM: normal metabolizer, IM: intermediate metabolizer, PM: poor metabolizer. * *p* < 0.05 in the univariate analysis. ** *p* < 0.05 in the univariate and multivariate analysis.

**Table 1 pharmaceutics-15-00404-t001:** Genotyped SNPs and alleles in which those SNPs are present.

Gene	Allele	SNPs Present in the Allele	Gene	Allele	SNPs Present in the Allele
*CYP1A2*	N/A	rs2069514	*CYP2D6*	*3	rs35742686
	N/A	rs762551		*4	rs3892097, rs1065852
	N/A	rs2470890		*6	rs5030655
*CYP2A6*	*9	rs28399433		*7	rs5030867
*CYP2B6*	*4	rs2279343		*8	rs5030865
	*5	rs3211371		*9	rs5030656
	*6	rs2279343, rs3745274		*10	rs1065852
	*7	rs2279343, rs3745274, rs3211371		*14	rs5030865
	*9	rs3745274		*17	rs28371706
*CYP2C8*	*2	rs11572103		*41	rs28371725
	*3	rs10509681	*CYP3A4*	*2	rs55785340
	*4	rs1058930		*6	rs46464389
*CYP2C9*	*2	rs1799853		*20	rs67666821
	*3	rs1057910		*22	rs35599367
*CYP2C19*	*2	rs4244285	*CYP3A5*	*3	rs776746
	*3	rs4986893		*6	rs10264272
	*4	rs28399504	*SLC22A1*	N/A	rs12208357
	*17	rs12248560		N/A	rs34059508
*ABCB1*	N/A	rs1045642 (C3435T)		N/A	rs72552763
	N/A	rs2032582 (G2677A/T)	*SLCO1B1*	*5	rs4149056
	N/A	rs1128503 (C1236T)		*15	rs4149056, rs2306283
*ABCC2*	N/A	rs2273697		*37	rs2306283
*ABCG2*	N/A	rs2231142		N/A	rs4149015
*UGT1A1 ^#^*	*80	rs887829			

SNP: single nucleotide polymorphism; ^#^
*UGT1A1* rs887829 was used as a surrogate predictor of *UGT1A1*28.*

**Table 2 pharmaceutics-15-00404-t002:** Demographic characteristics regarding sex, race and clinical trial.

	N	Age (Years)	Height (m)	Weight (kg)	BMI (kg/m^2^)
**Sex**					
Men	81	25.48 (5.06)	1.77 (0.07)	77.26 (10.00)	24.58 (2.76)
Women	79	27.79 (8.43)	1.63 (0.06) *	61.10 (7.85) *	23.08 (2.80) *
**Race**					
Caucasian	118	25.63 (6.00)	1.71 (0.09)	69.33 (11.73)	23.60 (2.92)
Latin-American	36	30.19 (9.10) *^1^	1.67 (0.11)	68.83 (13.43)	24.56 (2.72)
Other ^#^	6	24.67 (4.41)	1.71 (0.10)	71.08 (12.68)	24.14 (2.34)
**Clinical trial**					
A	22	26.05 (6.44)	1.70 (0.12)	66.85 (14.00)	22.85 (2.88)
B	20	23.90 (2.75)	1.72 (0.09)	68.37 (10.91)	23.09 (2.99)
C	4	30.00 (8.76)	1.70 (0.16)	77.93 (14.58)	26.85 (2.45)
D	25	27.64 (6.78)	1.67 (0.08)	65.90 (10.79)	23.68 (2.84)
E	16	25.56 (7.46)	1.67 (0.09)	64.73 (10.36)	23.17 (2.34)
F	31	26.81 (8.86)	1.71 (0.08)	72.16 (11.12)	24.57 (2.90)
G	25	27.64 (8.04)	1.72 (0.10)	71.45 (11.75)	24.17 (2.54)
H	18	27.43 (5.20)	1.72 (0.12)	72.29 (14.20)	24.33 (3.22)
**Total**	160	26.62 (7.01)	1.70 (0.10)	62.80 (12.09)	23.84 (2.87)

Data are shown as mean (standard deviation). BMI: body mass index. ^#^ Other: self-reported as Blacks, Asians or Arabs. * *p* < 0.05 compared to men. *^1^
*p* < 0.05 compared to Caucasians + healthy volunteers with other races.

**Table 3 pharmaceutics-15-00404-t003:** Pharmacokinetic parameters regarding sex, race, clinical trial and co-administered drug.

	N	AUC_72_/DW (h*ng*kg/mL*mg)	C_max_/DW (ng*kg/mL*mg)	t_max_ (h)	t_1/2_ (h)
**Sex**					
Men	81	2121.71 (467.12)	65.39 (12.52)	5.96 (1.29)	35.89 (6.48)
Women	79	2127.07 (520.90)	63.72 (13.44)	6.53 (1.58) *	33.87 (6.98)
**Race**					
Caucasian	118	2088.24 (514.93)	64.74 (13.25)	6.09 (1.46) *^1^	34.57 (7.52)
Latin-American	36	2225.91 (412.95)	63.76 (12.00)	6.72 (1.26)	35.81 (5.24)
Other	6	2225.25 (457.01)	66.09 (15.02)	6.42 (2.33)	35.65 (4.53)
**Clinical trial**					
A	22	1887.51 (397.95)	61.52 (12.27)	6.30 (1.32)	34.96 (7.83)
B	20	2377.38 (619.04)	71.67 (14.54)	6.85 (1.18)	40.19 (6.91)
C	4	2223.35 (480.40)	68.07 (12.55)	6.25 (0.50)	37.24 (5.24)
D	25	2075.88 (355.09)	63.62 (11.30)	6.72 (1.30)	33.07 (4.83)
E	16	2095.49 (327.75)	61.09 (8.87)	6.47 (1.26)	33.26 (5.64)
F	31	2397.11 (517.72)	70.20 (10.85)	5.94 (1.67)	35.61 (7.75)
G	25	1951.39 (500.31)	60.35 (15.01)	5.22 (1.58) *^2^	33.63 (7.83)
H	17	1965.30 (397.43)	59.94 (12.82)	6.59 (1.25)	32.79 (4.38)
**Co-administered drug**					
Atorvastatin	22	1887.51 (397.95) *^3^	61.52 (12.27)	6.30 (1.32)	34.96 (7.83)
Valsartan	20	2377.38 (619.04)	71.67 (14.54)	6.85 (1.18)	40.19 (6.91) *^3^
Valsartan + HTZ	45	2095.96 (350.46)	63.11 (10.54)	6.59 (1.23)	33.51 (5.18)
Olmesartan + HTZ	73	2143.91 (527.46)	64.43 (13.62)	5.84 (1.62)	34.28 (7.16)
**Total**	160	2124.35 (492.85)	64.56 (12.97)	6.24 (1.47)	34.89 (6.98)

HTZ: hydrochlorothiazide. Data are shown as mean (standard deviation). *: *p*_uv_ < 0.05 compared to men. *^1^: *p*_uv_ < 0.05 compared to Latin-Americans + healthy volunteers with other races. *^2^: *p*_uv_ < 0.05 compared to other clinical trials. *^3^: *p*_uv_ < 0.05 compared to the co-administration of other drugs. Underlined: *p*_mv_ < 0.05 compared to men, to Latin-Americans and healthy volunteers with other races, to other clinical trials or to other co-administered drugs, respectively.

**Table 4 pharmaceutics-15-00404-t004:** Pharmacokinetic parameters according to CYP3A4 phenotype and genotypes or phenotypes showing statistically significant associations in the exploratory step.

Genotype or Phenotype	N	AUC_72_/DW (h*ng*kg/mL*mg)	C_max_/DW (ng*kg/mL*mg)	t_max_ (h)	t_1/2_ (h)
**CYP2D6**					
UM	4	1971.00 (344.28)	61.43 (7.83)	5.63 (1.49)	35.73 (4.63)
NM	35	2106.89 (438.36)	63.78 (13.11)	6.19 (1.30)	34.01 (6.23)
IM	21	2166.75 (319.17)	66.92 (10.74)	6.12 (1.46)	36.13 (4.13)
PM	8	2307.66 (713.05)	66.64 (12.15)	5.94 (1.99)	40.45 (10.87) *^1^
***SLC22A1* rs34059508**					
G/G	66	2110.77 (428.73)	64.38 (11.95)	6.07 (1.39)	35.19 (6.51)
G/A	3	2638.62 (246.80) *^2^	72.35 (12.85)	7.17 (2.02)	40.69 (4.17)
**CYP3A4**					
NM	64	2128.17 (448.82)	64.82 (12.23)	6.09 (1.39)	35.23 (6.70)
IM	5	2231.76 (260.19)	63.53 (9.51)	6.40 (1.85)	38.00 (2.10)

Data are shown as mean (standard deviation). *^1^: *p_uv_* < 0.05 compared to UMs +NMs + IMs. *^2^
*p_uv_* < 0.05 compared to G/G genotype. Underlined: *p*_mv_ < 0.05 compared to UMs + NMs + IMs or compared to G/G genotype. No genetic information was available for one volunteer, as no sample could be recovered. The total number of volunteers for some genes is lower than 69 due to errors in the genotyping technique.

**Table 5 pharmaceutics-15-00404-t005:** Pharmacokinetic parameters regarding the genes analyzed in the confirmatory step.

Genotype or Phenotype	N	AUC_72_/DW (h*ng*kg/mL*mg)	C_max_/DW (ng*kg/mL*mg)	t_max_ (h)	t_1/2_ (h)
**CYP2D6**					
UM	13	1905.21 (291.68) *	60.59 (7.87)	5.89 (1.31)	33.68 (4.62)
NM	87	2123.30 (497.73)	64.41 (13.79)	6.36 (1.36)	34.64 (7.54)
IM	48	2164.35 (485.10)	66.08 (12.90)	6.18 (1.66)	34.84 (5.56)
PM	8	2307.66 (713.05)	66.64 (12.15)	5.94 (1.99)	40.45 (10.87) *^1^
** *SLC22A1* ** **rs34059508**					
G/G	152	2111.66 (492.69)	64.44 (13.02)	6.20 (1.46)	34.78 (7.07)
G/A	6	2442.42 (391.44) *^2^	69.05 (12.78)	7.25 (1.75)	37.63 (5.47)
**CYP3A4**					
NM	144	2111.68 (483.74)	64.51 (13.02)	6.19 (1.48)	34.87 (6.98)
IM	14	2316.87 (570.00)	66.35 (13.22)	6.82 (1.32)	35.12 (7.69)

Data are shown as mean (standard deviation). * *p*_uv_ < 0.05 compared to NMs + IMs + PMs. *^1^
*p*_uv_ < 0.05 compared to IMs + NMs + UMs. *^2^
*p*_uv_ < 0.05 compared to G/G genotype. Underlined: *p*_mv_ < 0.05 compared to NMs + IMs + PMs. No genetic information was available for one volunteer, as no sample could be recovered. The total number of volunteers for some genes is lower than 159 due to errors in the genotyping technique.

**Table 6 pharmaceutics-15-00404-t006:** Significant differences in the incidence of ADRs according to clinical trial design, demographic characteristics and pharmacogenetic variables.

Variable	Category	N	ADR	Volunteers Affected	Significance
Sex	Men	81	Any ADR	26 (32.1%)	*p*_uv_ = 0.024, *p*_mv_ = 0.012, log OR = 2.327
	Women	79		40 (50.6%)	
Race	Caucasian	118		41 (34.7%)	*p*_uv_ = 0.006, *p*_mv_ = 0.031, log OR = 2.306
	Latin-American	36		20 (55.6%)	
	Other	6		5 (83.3%)	
Clinical trial	A	22		6 (27.3%)	*p*_uv_ = 0.014
	B	20		3 (15.0%)	
	C	4		0 (0.0%)	
	D	25		14 (56.0%)	
	E	16		7 (43.8%)	
	F	31		16 (51.6%)	
	G	25		14 (56.0%)	
	H	17		6 (35.3%)	
Co-administered drug	Atorvastatin	22		6 (27.3%)	*p*_uv_ = 0.011
	Valsartan	20		3 (15%)	
	Valsartan + HTZ	45		21 (46.7%)	
	Olmesartan + HTZ	73		36 (49.3%)	
Race	Caucasian	118	Headache	29 (24.6%)	*p*_uv_ = 0.002, *p*_mv_ = 0.001, log OR = 3.300
	Latin-American	36		17 (47.2%)	
	Other	6		5 (83.3%)	
*SLC22A1* rs34059508	G/G	153	Thoracic pain	0 (0.0%)	*p*_uv_ = 0.038
	G/A	6		1 (16.7%)	
*SLC22A1* rs34059508	G/G	153	Dizziness	7 (4.6%)	*p*_uv_ = 0.038, *p*_mv_ = 0.014, log OR = 10.975
	G/A	6		2 (33.3%)	

ADR: adverse drug reaction, HTZ: hydrochlorothiazide.

## Data Availability

Data belong to the clinical trials’ sponsors and may be accessible upon reasonable request to the corresponding authors.

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
