# Peer review of "Genetic Variation in CYP2D6 and SLC22A1 Affects Amlodipine Pharmacokinetics and Safety"

_pharmaceutics, 2023, doi:10.3390/pharmaceutics15020404_

Round 1

Reviewer 1 Report

This manuscript presents the results of a clinical study on pharmacokinetics of amlodipine in 160 healthy volunteers and their correlation with genetic polymorphisms.

The paper is correctly written. There are big tables, not too easy to read, but the results seem well interpreted. The methodology used uses only qPCR data on the diverse human CYP and transporters and a HPLC/MS technique done by a commercial laboratory for amlodipine analysis.The statistical data are correctly given.

Only two genotypes seem to emerge: CYP2D6 ultra-metabolisers have a slightly lower AUC that the general population and Poor-metabolisers a prolongated half life of the drug.

And the cationic transporter OCT1, SLC22A1  Patients with G/A have also a higher AUC and prolongated half-life. They had adverse reactions.

It would be nice to look in some normal volunteers of quindine a strong CY2D6 inhibitor is able to modify the clearance and/or half-life of the drug in the same volunteer.

But I checked if this interaction is known. On that page

https://www.pdr.net/drug-summary/Quinidine-Sulfate-Extended-Release-Tablets-quinidine-sulfate-790

There is a amlodipine–quinidine paragraph with no known problem. Interaction with CYP3A4 inhibitors is also reported in the literature (azole drugs).

I find the tables difficult to read. However this is probably the best way to report the data. Perhaps two diagrams showing the AUC  and the half-life in fonction of the genetic parameters tested could have been illustrative. (with error bars and significance). I suggest to try.

Amlodipine is a diester and primary amine.  Amine are often substrates for CYP2D6.

In conclusion I think that the paper is correctly written and perhaps interesting. It could be published with minor corrections.

Reviewer 2 Report

The authors submitted a research article in which they elucidated a role of SNPS in amlo degradation in connection with its plausible pharmacological effects in Latin Americans vs other races. They included 216  and 160 healthy volunteers in the study and found strong relationship between amlodipine and CYP2D6 and SLC22A1. Although these findings are intriguing, I would like to put forward several comments to discuss.

1. The authors indicated that CYP2D6 and SLC22A1 are potential pharmacogenetic biomarkers in amlodipine treatment, whereas there was not performed a discovery of others with GWAS. Please, add a short comment and add it to a section Study limitations.

2. Please, add a brief comment at the end of the section Discussion whether some clinically significant changes in AUCs of amlo in Latin Americans might rlevant in the same manner in other patients with hypertension and comorbidities, such as diabets, CKD, etc,
